# Effects of Bias Voltages on the Structural, Mechanical and Oxidation Resistance Properties of Cr–Si–N Nanocomposite Coatings

**Yanxiong Xiang, Lingling Huang and Changwei Zou \***

School of Physics Science & Technology, Lingnan Normal University, ZhanJiang 524048, China;
yanxiongxiang@163.com (Y.X.); 18558641895@163.com (L.H.)

\* Correspondence: Zoucw@lingnan.edu.cn; Tel./Fax: +86-759-3183260

**Abstract:** Cr–Si–N nanocomposite coatings were deposited by multi-arc ion plating under different bias voltages. The influences of bias voltage on composition, microstructure, surface morphology and mechanical properties of Cr–Si–N nanocomposite coatings were investigated in detail. The HR-TEM, XRD, and XPS results confirmed the formation of nanocomposite structure of nanocrystalline of CrN embedded into the amorphous phase of $Si_3N_4$. The particle radius of CrN can be calculated from the half-width of the diffraction peak of CrN (200) and the value was about 20–60 nm. In addition, no diffraction peaks of $CrSi_2$, $Cr_3Si$, or $Si_3N_4$ were found in all the Cr–Si–N coatings. With the increasing of bias voltages from 0 to −200 V, the number and size of large droplets on the coating surface decreased, and the growth mode of the coatings changed from loose to dense. However, with the increasing of bias voltages from 0 to −200 V, the micro-hardness of the coatings increased and then decreased, reaching its maximum value at negative bias voltages of 100 V. It was found that the friction coefficient of Cr–Si–N coatings is almost the same except for the Cr–Si–N coatings deposited under bias voltage of 0 V. When the oxidation temperature was at 800 °C, the Cr–Si–N coating was only partially oxidized. However, with the increase of oxidation temperature to 1200 °C, the surface of the coating was completely covered by the oxide generated. The results showed that the bias voltages used in multi-arc ion plating had effects on the structure, mechanical, and high temperature oxidation resistance properties of Cr–Si–N nanocomposite coatings.

**Keywords:** Cr–Si–N; nanocomposite coating; MULTI-arc ion plating; bias voltage; high-temperature oxidation resistance

## 1. Introduction

Because of its excellent properties, such as excellent oxidation resistance at high temperature, higher hardness, and good corrosion resistance, CrN has become a potential protective material [1–3]. CrN based coatings deposited by physical vapor deposition (PVD) methods, such as magnetron sputtering, multi-arc ion plating, and ion beam sputtering have been widely studied [4–6]. However, the main shortcoming of CrN coating is its hardness of about 18 GPa, which is lower than that of TiN (21–23 GPa) [7]. Nanocomposite nitride coatings (recognized as nc-MeN/a-$Si_3N_4$) consist of transition-metal nitride nanocrystallite surrounded by amorphous matrix [8]. The modification of the CrN lattice, by the addition of Si atoms, causes enhanced hardness of the structure due to the solid solution hardening mechanism. Moreover, the silicon nitride phase was oxidized into the $SiO_2$ phase which would prevent the inward diffusion of oxygen into the coating, and eventually improved the oxidation resistance of the Cr–Si–N coating [9]. In addition to maintaining the excellent high temperature oxidation resistance properties of CrN, the hardness, wear resistance and other mechanical

properties of the Cr–Si–N coatings were effectively improved, which laid a good foundation for the wider application of Cr–Si–N in industrial production [10,11].

One of the most obvious characteristics of multi-arc ion plating is its higher ionization rate and deposition efficiency in the preparation of tool coatings when compared with magnetron sputtering. Bias voltage is an important deposition parameter in the technological process of multi-arc ion plating. Applying a negative bias voltage to the substrate has been shown to increase the kinetic energy of the bombarding positive ions [12]. On the one hand, the energy obtained from the electric field provided by the bias voltage can promote the growth of grain on the coating surface [13]. On the other hand, with the increase of surface energy, the toughness and compactness of coatings are optimized to a great extent. However, in order to achieve the positive effect for the resulting coatings, the bias voltages must be controlled in a suitable range. During deposition process, too large or too small of bias will have a negative impact on the mechanical properties of the coatings. Parra [14] found that the hardness and friction properties of CrN increased with the increase of bias voltage. However, with the increase of substrate bias, the number of large particles on the surface and deposition rate of the coatings decreased, which was attributed to the ion bombardment and re-sputtering effects [15]. It has previously been reported that increasing the energy of incident ions generates a larger number of defects on the surface of growing films, which provide a larger number of nucleation sites and a corresponding increase in the number of grains [16,17]. Bias voltage was also reported to have considerable influence on the residual stress and grain size of hard coatings [18].

Up to now, the effects of bias on the structure and deposition process of the coatings are still unclear. More importantly, there are few studies reported about the relationship between hardness mechanism, high temperature elemental diffusion and oxidation resistance properties of Cr–Si–N coatings deposited under different bias voltages. In this paper, the influence of bias voltages on the composition, structure, and high-temperature oxidation resistance mechanism of the Cr–Si–N nanocomposite coatings is explained.

## 2. Experimental Details

### 2.1. Coating Deposition

The single-crystal silicon (111), WC–Co cemented carbide and single-crystal NaCl (100) with sizes of $1 \times 1$ cm$^2$ were selected as substrates for deposition. The single-crystal silicon (111) was used for *X*-ray Powder Diffraction (XRD) and *X*-ray photoelectron spectroscopy (XPS) tests, the WC–Co cemented carbide was used for mechanical tests, and the single-crystal NaCl was used for transmission electron microscopy (TEM) tests. Before deposition, the substrates were ultrasonic cleaned in acetone and anhydrous ethanol for 20 min, and then dried with nitrogen before being put into the deposition chamber. The diameter of the chamber was 800 mm and the height was 600 mm. The cathode Cr target and the CrSi target were symmetrically installed on the wall of the barrel-shaped vacuum chamber. The pulse DC power supply (Pinnacle Plus, Colorado, AE, USA) was used for bias voltage. The distance between the substrate and the multi-arc targets was set at 360 mm. The substrates temperature was heated to 300 °C and the background vacuum was pumped to $4.0 \times 10^{-3}$ Pa. The Cr metal bonding layer was firstly deposited. The working pressure of Ar was set to 1.0 Pa, duty cycle was adjusted to 80%, substrate bias was maintained at −160 V, multi-arc target current was controlled at 80 A, and coating deposition time was maintained for 5 min. Then, the N$_2$ flow controller was modulated and the working pressure was controlled to 1.0 Pa, the bias voltage was reduced to −140 V, the multi-arc target current was maintained at 80 A and the CrN transition layer was deposited for 5 min. Finally, the Cr–Si–N coating was deposited. The working pressure and duty cycle were controlled at 1.0 Pa and 80%, respectively. The current of multi-arc Cr targets was fixed at 70 A, and the deposition time of the coating was kept for 50 min. Five different groups of Cr–Si–N coatings with substrate biases of 0, −50, −100, −150, and −200 V were deposited.

*2.2. Characterization*

The electron probe micro-analyzer (EPMA, EPMA 1600, Shimadzu, Kyoto, Japan) was used to analyze the composition of the coatings. The *X*-ray powder diffractometer (XRD, X'Pert-MPD System, PHILIPS, Amsterdam, The Netherlands) was used to detect the phase of the Cr–Si–N coatings with the step length of 0.02°/s. *X*-ray electron spectroscopy (XPS, PHI X-tool, ULVAC-PHI, Inc., Chigasaki, Japan) was used to analyze the chemical state of elements in the coatings. The working voltage was set at 15 kV, and the working current was kept at 50 A. For the data analysis, C 1s (E = 284.6 eV) was used to calibrate the spectral data and XPS peak 4.1 software was used for peak segmentation and fitting of the spectral data. The field emission scanning electron microscope (MERLIN, ZEISS, Jena, Germany) was used to observe the surface and cross-sectional morphology of coatings. The microstructure of the Cr–Si–N coatings were further investigated by transmission electron microscopy (TEM: JEOL JEM 2010F, Toyoshima, Japan) working at 200 kV. The surface morphology and root mean square (RMS) roughness were investigated using a Shimadzu SPM-9500J3 atomic force microscope (AFM, SHIMADZU, Kyoto, Japan) operated in the tapping mode with a measuring area of $3 \times 3$ mm$^2$. The hardness of the coatings was measured by the micro-hardness tester (ShangHai TaiMing Optical Instrument Co., Ltd., Shanghai, China). The loading time was kept at 20 s and the results were averaged over 10 measurements. The friction coefficient and wear performance of the coatings were tested by multifunctional surface performance tester (CFT-1 type). The friction material was Si$_3$N$_4$ ceramic ball with diameter of 4 mm. The rotating speed was 0.3 m/s, and the applied load was 80 g. The temperature and relative humidity were basically controlled at 20 °C and 75%, respectively. The high-temperature vacuum tube furnace was used for high-temperature oxidation experiment. The heating rate was 10 °C/min, and the temperature was maintained at 800 and 1200 °C for 2 h. The oxidation resistance property was evaluated by the change of elemental diffusion and morphology of the coatings before and after oxidation.

## 3. Results and Discussion

*3.1. Characteristics of Coatings*

EPMA characterization of Cr–Si–N coatings prepared under different bias voltages was shown in Figure 1. With the increase of bias voltage, there are two obvious forms of changes for the element content in the coatings. Firstly, with the increase of bias voltage from 0 to −50 V, the content of Cr increased suddenly, while the content of Si decreased sharply. This was mainly due to the emergence of bias voltage which led to a sudden electric field between the multi-arc target and the substrates. Compared with Si, the Cr ions were easier to combine with N. Therefore, the content of Cr increased while the content of Si decreased. Another reason may be the different ionization degrees of the elements involved, and the changes in plasma due to the presence of bias voltage. In addition, when the bias voltage was increased from −50 to −200 V, the atomic percentages of Cr and Si contents were close to the atomic percentages of the targets.

As can be seen from Figure 2, all the Cr–Si–N coatings had diffraction peaks of CrN (111) and CrN (200). However, when the bias voltage was increased to −150 V, CrN (220) diffraction peaks appeared. With the increase of bias voltage, the internal stress of the coating increased and the growth direction of the coating would change to (220) surface [19]. The tendency toward a specific orientation can be explained by strain and surface energy. Coatings of greater thicknesses tend toward a (111) orientation, due to the effects of strain energy [18]. The particle radius of CrN can be calculated from the half-width of the diffraction peak of CrN (200) and the value was about 20–60 nm. In addition, no diffraction peaks of CrSi$_2$, Cr$_3$Si or Si$_3$N$_4$ were found in all the Cr–Si–N coatings. It can be inferred that the Si$_3$N$_4$ phase in the Cr–Si–N coating prepared in this experiment probably existed in the amorphous form.

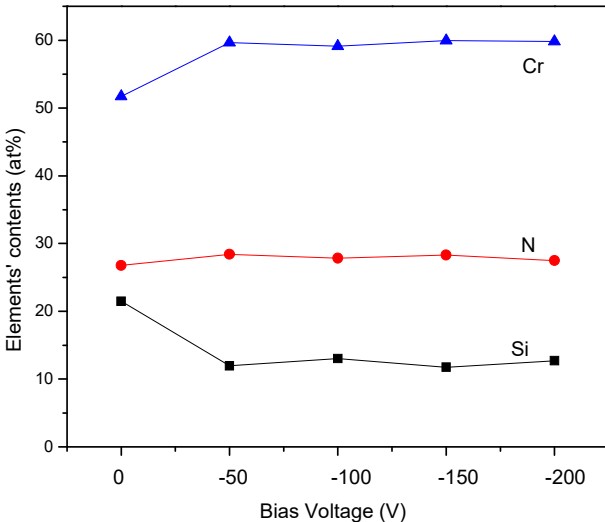

**Figure 1.** Element contents in Cr–Si–N coatings deposited under different bias voltages.

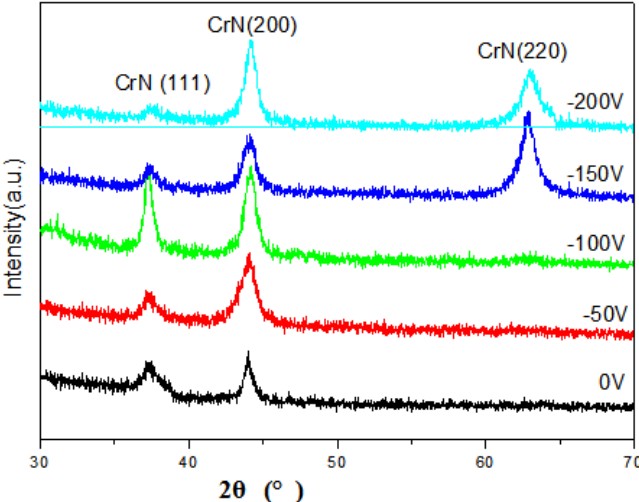

**Figure 2.** XRD spectra of Cr–Si–N coatings deposited under different bias voltages.

XPS spectra of Si 2p and N 1s for Cr–Si–N coatings deposited at −100 V were shown in Figure 3. The binding energy of Si 2p peak was centered at 101.5 eV, which was well consistent with the peak of Si–N bond in $Si_3N_4$ [20]. In addition, as can be seen from the Gauss fitting results of N 1s shown in Figure 3b, the peak with binding energy of 396.7 eV corresponded to the compound CrN, while the peak with binding energy of 397.9 eV coincided with the peak of Si–N in $Si_3N_4$ [21]. The peak area of CrN was larger than that of $Si_3N_4$ in the fitting results, which indicated that the main phase in the Cr–Si–N coatings was CrN phase. The binding energy peak of 397.9 eV in N 1s spectra further confirmed the existence of $Si_3N_4$ in Cr–Si–N coatings. When the bias voltage is −100 V, the peak of Cr $2p_{3/2}$ is centered at 573.2 eV. Compared with the standard value table of XPS, it can be found that the bond energies of CrN and $Cr_2O_3$ are located at 575.27 and 577.38 eV, respectively [20,21]. Therefore, CrN and $Cr_2O_3$ may exist simultaneously in the Cr–Si–N coatings. However, no $Si_3N_4$ diffraction peak was found in XRD spectra, but there were CrN diffraction peaks. Therefore, the results of XRD and XPS indicated that the structure of the Cr–Si–N coatings was composed of nanocrystalline CrN embedded in amorphous $Si_3N_4$ matrix.

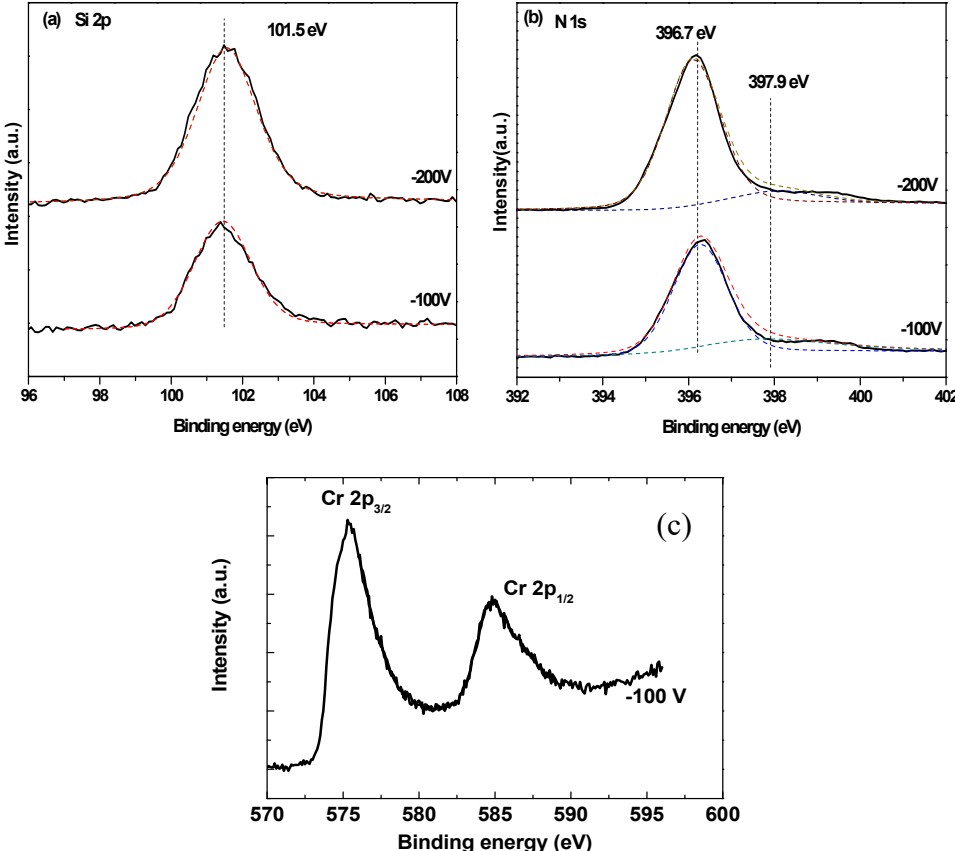

**Figure 3.** Si 2p (**a**), N 1s (**b**) and Cr 2p (**c**) spectra of Cr–Si–N coatings deposited under different bias voltages.

In order to investigate the structure details, TEM was conducted, together with the corresponding HR-TEM and FFT (fast Fourier transform) patterns of the Cr–Si–N coatings deposited under bias voltages of −100 V. As shown in Figure 4a, the Cr–Si–N coating exhibits a typical nanocomposite structure, in which nanocrystalline compounds with diameters of 30–50 nm are embedded in the amorphous matrix. The HR-TEM observations shown in Figure 4b,c confirm the formation of nanocomposite structure of nanocrystalline of nc-CrN embedded into the amorphous phase of $\alpha$-Si$_3$N$_4$ as explored by XRD and XPS. The HR-TEM image shown in Figure 4b with corresponding FFT pattern at area "A" demonstrates nanocrystalline separated by disordered grain boundaries.

Figure 5 shows the surface and cross-sectional morphology of Cr–Si–N coatings deposited under different bias voltages. It can be clearly seen that large droplet-like particles of different numbers and sizes are distributed on the surface of all Cr–Si–N coatings. This is mainly due to the fact that when arc multi-arc target is evaporated, large Cr metal droplets with initial momentum are deposited on the substrates. However, with the increase of substrate bias voltage from 0 to −200 V, the number and size of large droplets on the surface of the coating decrease, and the RMS roughness of the coating surface decrease from 84 to 57 nm. On the one hand, these charged ions will deposit on the surface of the substrates to form a film under the action of electric field. On the other hand, because the large droplets are attached to the surface of the substrate only by inertial force, the binding force between the droplets and the substrate is relatively poor, which will lead to cracks in the coating due to loose bonding. The energetically accelerated charged ions will bombard the large droplets on the surface of the coating, and the number and size of large droplets decrease with the increase of bias voltage. The number and size of some micro-pits distributed on the surface of the coating increase gradually. The reason may be due to larger stress of the coatings, higher energy of droplets and smaller sticking effect, re-sputtering, heating or surface charge accumulation etc.

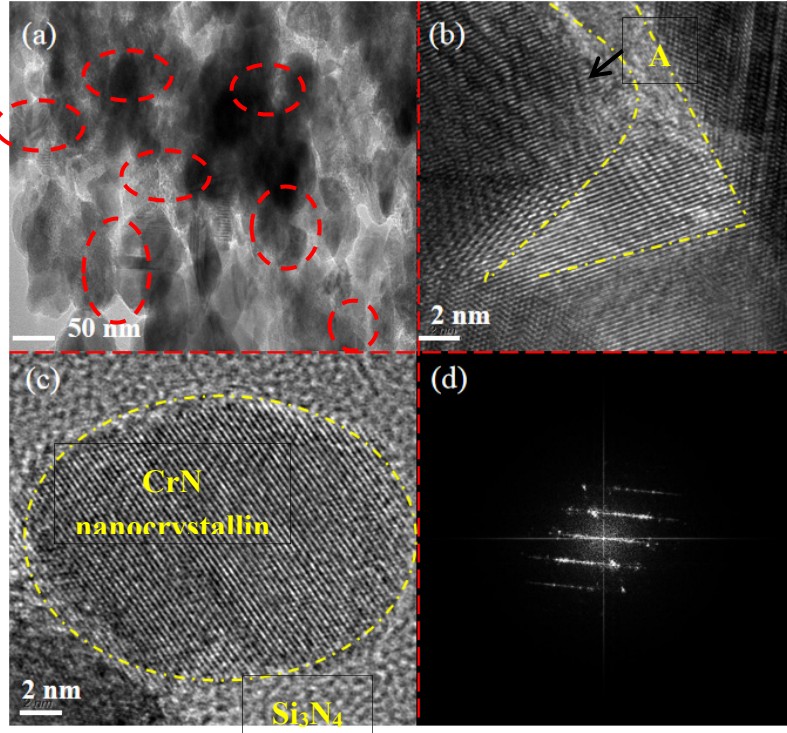

**Figure 4.** TEM (**a**), HR-TEM (**b**,**c**) and FFT (fast Fourier transform) images (**d**) of Cr–Si–N coatings deposited under bias voltages of −100 V.

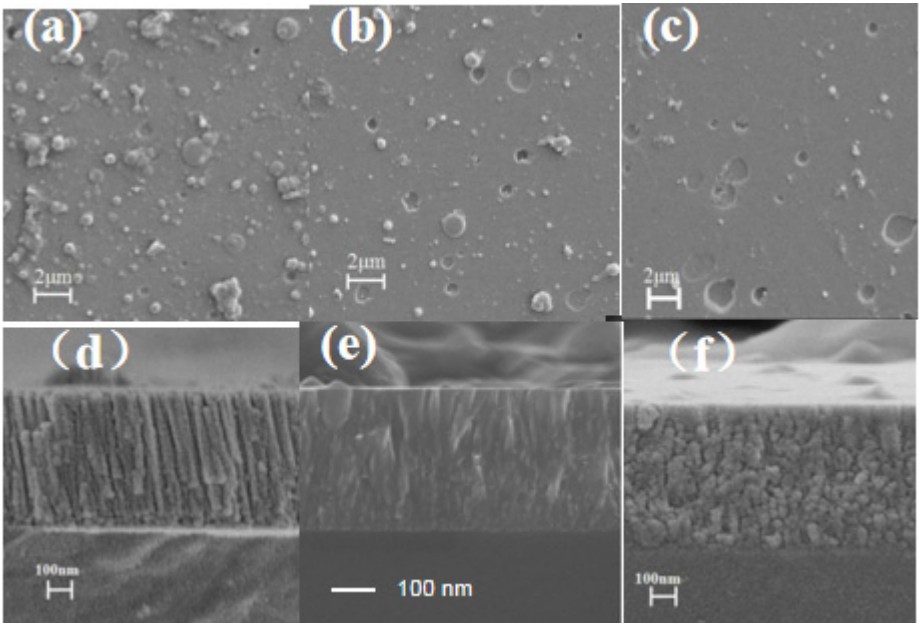

**Figure 5.** Surface (**a**–**c**) and cross-sectional (**d**–**f**) images of Cr–Si–N coatings deposited under bias voltages of 0 V (**a**,**d**), −100 V (**b**,**e**), and −200 V (**c**,**f**), respectively.

From the cross-sectional images shown in Figure 5, it can be seen that the bias voltage has a significant effect on the growth mode of the Cr–Si–N coatings. When the bias voltage is set at 0 V, the growth mode of Cr–Si–N coatings is consist of coarse and loose columnar structure. However, when the bias voltage is increased to −100 V, the cross-sectional image of the coatings shows that the structure of the coatings is compact and the surface grain becomes small. However, when the bias

voltage is further increased, the grains of the coating become loose and slightly cracked. When the bias voltage is not applied, the diffusion of the grain boundary is finally prevented by the "thickness effect" of the coatings, resulting in the formation of the columnar crystal structure at the grain boundary [22]. When the bias voltage is increased from 0 to −100 V, the bombardment effect enhanced and the mobility of particles and the grains become more compact. However, the further increase of bias makes the ability of bombarding ions too high, which is why the grain boundary appears again in the coatings deposited under bias voltages of −200 V.

　　　Figure 6 shows the deposition rate of Cr–Si–N coatings prepared under different bias voltages calculated according to the SEM cross-sectional images. The deposition rate of the coatings firstly increases and then decreases with the increase of bias. The increased flux and mobility of the bombardment ions would increase the number of ions deposited on substrates, which further led to the increasing of thickness of the coatings. However, when the bias voltage was too high, it would lead to the occurrence of a re-sputtering phenomenon [23].

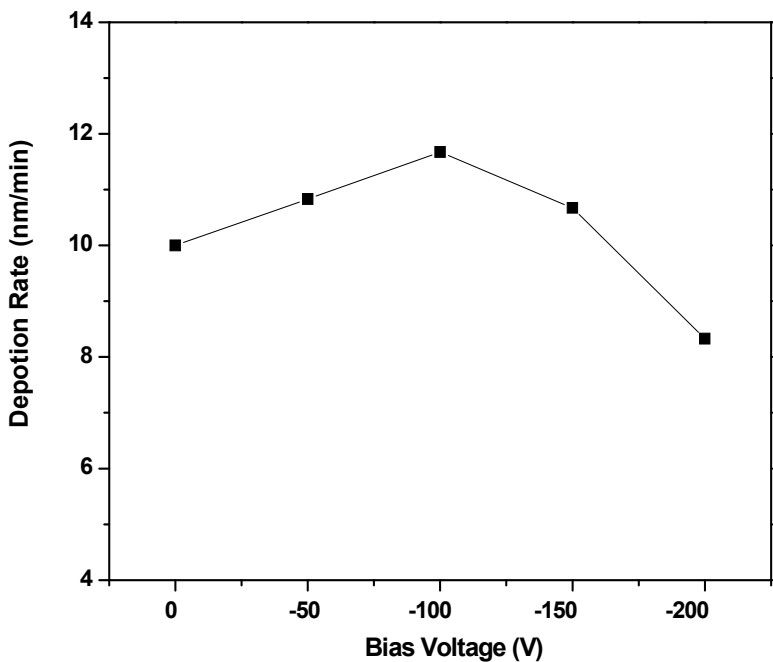

**Figure 6.** Deposition rates of Cr–Si–N coatings deposited under different bias voltages.

### 3.2. Mechanical Properties

　　　Figure 7 shows the hardness of the Cr–Si–N coatings deposited under different bias voltages. The hardness of Cr–Si–N coatings increases and then decreases with the increase of bias voltages. With the increase of substrate bias from 0 to −100 V, the energy of bombardment ion increases rapidly. However, when the substrate bias exceeds −100 V, the phenomenon of re-sputtering occurs, which not only increases the internal stress of the coatings, but also causes more defects on the surface of the coatings. Biswas et al. [23] reported that higher bias voltages tend to improve the density of thin films by enhancing the intensity of ion bombardment. However, excessively high bias voltages can compromise the mechanical properties. Lomellon et al. [24] reported an increase in the hardness of AlCrN coatings from 30 to 50 GPa after the bias voltage was increased from 0 to −150 V. In addition, no super hardness greater than 4000 HV appeared in our CrSiN coatings, which might be due to the fact that clear columnar grain boundaries were formed in the coatings and they could act as sites for cracking [8].

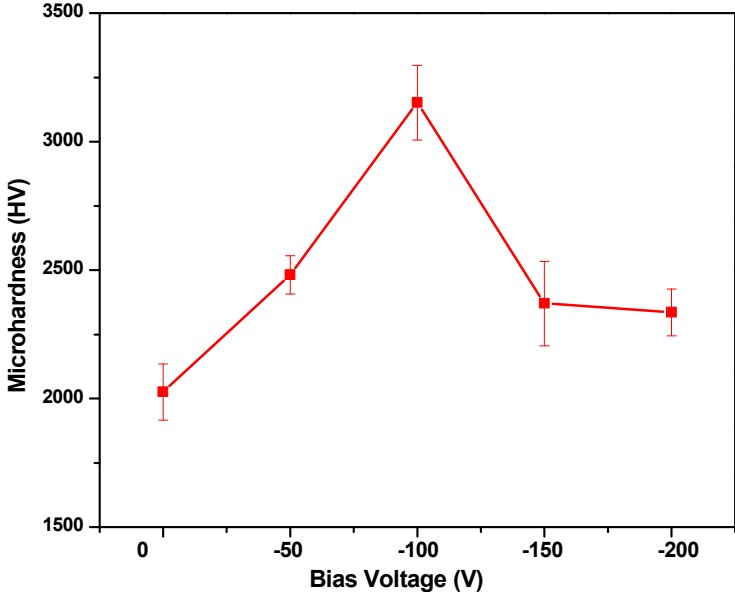

**Figure 7.** Micro-hardness values of Cr–Si–N coatings deposited under different bias voltages.

Figure 8 shows the friction coefficient of Cr–Si–N coatings deposited under different bias voltages. It is found that the friction coefficient of Cr–Si–N coatings is almost the same except for the Cr–Si–N coatings deposited under bias voltage of 0 V. This phenomenon may be related to different Si contents in the Cr–Si–N coatings. As has been shown in Figure 1, when the substrate bias is not applied, the content of Si in the Cr–Si–N coating is about 20 at.%. However, when the substrate bias is applied, the content of Si in the coatings remains basically unchanged. It is believed that the reduction of friction coefficient is caused by the reaction between Si in the coatings and water in the atmosphere [25]. The following reactions may occur on the surface of the coatings during wear process:

$$Si_3N_4 + 6H_2O \rightarrow 3SiO_2 + 4NH_3 \tag{1}$$

$$SiO_2 + H_2O \rightarrow Si(OH)_4 \tag{2}$$

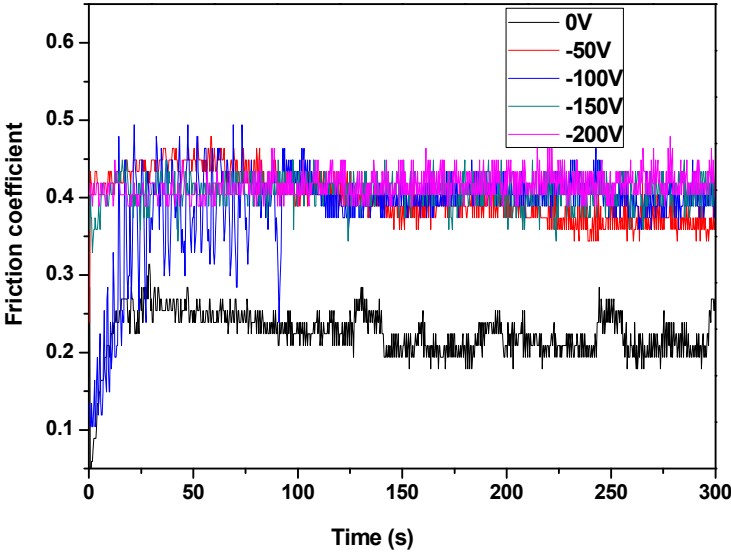

**Figure 8.** Friction coefficient of Cr–Si–N coatings deposited under different bias voltages.

The formed silicon hydroxide is considered as the protective layer on the coating surface, which can play a lubricating role, which effectively reduces the friction coefficient of the coatings. In addition, the tribological properties of the Cr–Si–N coatings are also related to hardness of the coating, surface roughness, and possible formation of a tribolayer from their residues.

### 3.3. High-Temperature Oxidation Resistance Properties

The experimental results show that the Cr–Si–N coatings prepared under bias voltage of −100 V has excellent structure and mechanical properties. Therefore, the high temperature oxidation resistance property of the Cr–Si–N coatings prepared under this parameter is studied. From the XRD spectra shown in Figure 9, it can be seen that there are only diffraction peaks of CrN in the as-deposited Cr–Si–N coatings [26].

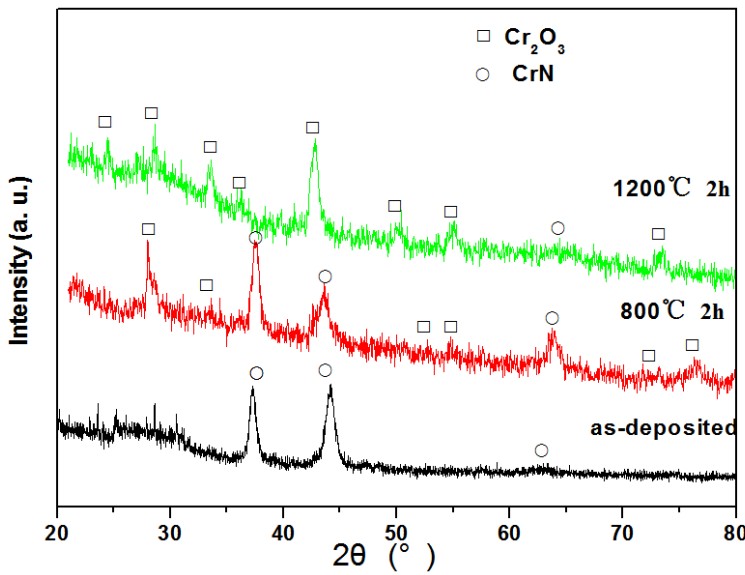

**Figure 9.** XRD spectra of Cr–Si–N coatings before and after oxidation.

When the coatings are oxidized at 800 °C in air for 2 h, there are still diffraction peaks of CrN phase in the coatings although several weak peaks of $Cr_2O_3$ appear [27–30]. It has been found that the oxidation of CrN coatings was very serious between 600 and 700 °C. It can be seen that the oxidation performance of the Cr–Si–N nanocomposite coatings formed by Si doping is indeed improved compared with that of CrN. However, when the Cr–Si–N coating was oxidized at 1200 °C in air for 2 h, the diffraction peaks of CrN phase were disappeared. The disappearance of the diffraction peaks of CrN phase indicated that oxygen diffused inward and Cr element diffused outward continuously in the oxidation process. The following reactions may occur:

$$2CrN \text{ (s)} + O_2 \text{ (g)} = Cr_2O_3 \text{ (s)} + N_2 \text{ (g)} \tag{3}$$

A large number of Cr oxides were produced in the coatings, and the N element was constantly volatilized into the air in the form of nitrogen. There was no diffraction peak of $SiO_2$ in the XRD spectra of the oxidized Cr–Si–N coatings. The possible reasons for this phenomenon may be that the affinity potential energy between Si and O was lower than that of Cr [29].

As shown in Figure 10, during the progress of high temperature oxidation, there were some large bulk particles on the surface of the coatings, which were the oxide particles of $Cr_2O_3$. When the oxidation temperature was set at 800 °C, the Cr–Si–N coating was only partially oxidized. However, with the increase of oxidation temperature to 1200 °C, the surface of the coating was completely covered by the oxide generated. As can be seen from the cross-sectional morphology, at oxidation temperature

of 800 °C, besides the roughness of the coatings due to the covering of oxide particles, the internal structure of the coatings at this time was still very compact, which indicated that the Cr–Si–N coatings can effectively withstand 800 °C oxidation. However, when the oxidation temperature increased to 1200 °C, serious internal distortion took place in the coating. Moreover, many obvious cracks appeared between the coating and the substrates, and the coating began to fall off. This phenomenon may be caused by the different thermal expansion coefficients of coatings and substrates during the high temperature oxidation process.

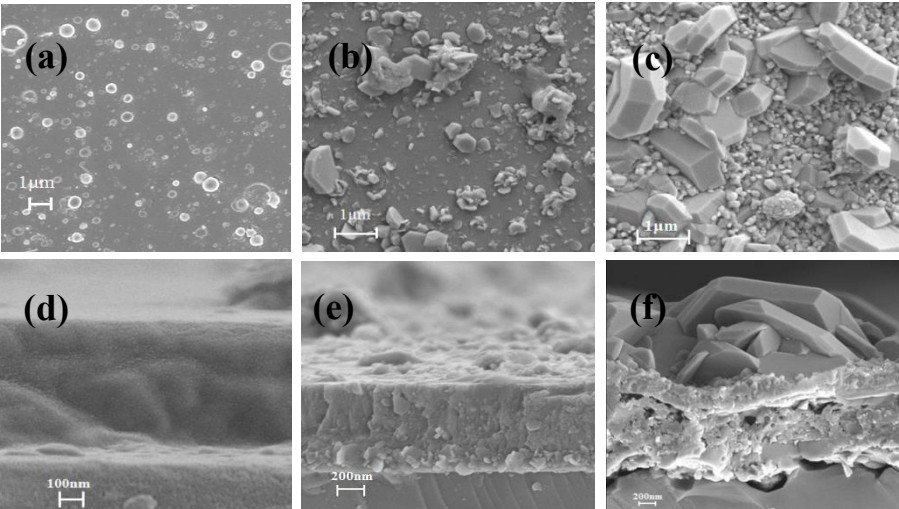

**Figure 10.** Surface (**a**–**c**) and cross-sectional (**d**–**f**) images of Cr–Si–N coatings before (**a**,**d**) and after oxidation at 800 °C (**b**,**d**) and 1200 °C (**c**,**f**).

## 4. Conclusions

Cr–Si–N nanocomposite coatings were deposited by multi-arc ion plating under different bias voltages. The number and size of large droplets on the surface of the coatings decreased when the bias voltage was increased from 0 to −200 V. When the bias voltage increased to −100 V, the cross-sectional image of the coating showed that the structure of the coating was compact and the grain size was decreased. The number and size of some micro-pits distributed on the surface of the coating increase gradually. The reason for this may be due to the larger stress of the coatings that support the droplets, higher energy of droplets, and smaller sticking effect, re-sputtering, heating, or surface charge accumulation. The HR-TEM, XRD, and XPS results confirmed the formation of nanocomposite structure of nanocrystalline of CrN embedded into the amorphous phase of $Si_3N_4$. The micro-hardness of the coating increased and then decreased with the increase of bias voltage from 0 to −200 V. The formed silicon hydroxide is considered as the protective layer on the coating surface, which can play a lubricating role, which effectively reduces the friction coefficient of the coatings. In addition, the tribological properties of the Cr–Si–N coatings were also related to hardness of the coating, surface roughness, and possible formation of a tribolayer from their residues. The coatings with the best hardness value deposited under bias voltage of −100 V can resist 800 °C and 2 h high temperature oxidation in air.

**Author Contributions:** C.Z. designed the experiments; Y.X. performed the experiments; L.H. and Y.X. analyzed the data. All authors have read and agreed to the published version of the manuscript.

**Funding:** This work was supported by major projects of basic and application research in Guangdong Province (2017KZDXM055), Special fund for science and technology innovation strategy of Guangdong Province (2018A03015), Natural Science Foundation of Guangdong Province (2018A030307027) and Zhanjiang Science and Technology plan (2018A02010).

**Conflicts of Interest:** The authors declare no conflict of interest.

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
