# Peer review of "Effects of Bias Voltages on the Structural, Mechanical and Oxidation Resistance Properties of Cr–Si–N Nanocomposite Coatings"

_coatings, doi:10.3390/coatings10080796_

Round 1
Reviewer 1 Report
The manuscript is fairly well written and shows a complete analysis of the effect of bias voltage to the properties of the Cr-Si-N composite coatings. The figures are well done a and clear too. However, there are some issues that should revised before publishing the manuscript.
1) 1. Introduction: could the motivation of the work be written out more clearly. E.g. what are the uses and applications of these CrN based protective coatings? And what are the advantages of the multi-arc deposition technique compared to the other PVD methods, especially magnetron sputtering? I would also suggest to add more relevant references here, as e.g. the refs [7] and [8] are not relevant to this context ([7] Bi et al. does not consider PVD techniques and [8] Rojas et al. does not even mention the word "bias" in the article.)
2) Line 40-41 What the authors mean by saying "..., the deposition chamber is filled with a large number of plasma" ? Should this be rewritten.
3) 2.1 Coating deposition: (i) What was the size of the substrates? (ii) What was the Si source, a solid Si target? A more detailed description, or at least a citation to such desrciption would be appreciated. (iii) What was the thickness of the Cr and CrN binding/transition layers?
4) Line 67 "Deposition room" should this be "deposition chamber"
5) 2.2. Characterization: (i) The authors should check and complete the correct experimental/instrument details. (ii) For XPS calibration C 1s (E= 286.4ev) typically a lower energy is used for adventitious carbon, 284.8-285, why such as high energy was chosen this time?
6) Results and discussion. In the experimental details it was said that the films were deposited on 3 different substrates, Si, WC-Co cemented carbide and NaCl. What was the significance of the different substrates to the film characteristics? It is not mentioned anywhere in the results, what is the substrate of the films in each analysis. The authors should also mention the substrate in each of the figure captions.
7) The EMPA results show that the N content in the films is only ca. 28%. This contradicts the authors' discussion of the film composition being nanocrystalline CrN and Si3N4 as the stoichiometry does not match with the measured composition. Could the authors elaborate this discrepancy?
8) XRD results. (i) What is the substrate? (ii) What is the reference card used for CrN identification. Or a literature reference? the cited ref [11] Carvalho et al. is not relevant here as it does not consider Cr-based materials.
9) XPS results. The fitting in the N 1s peak does not look correct. Despite the background fit not being visible, it is still obvious that the claimed peak corresponding to be Si-N is not there at 397.7, but a lesser peak appears at higher binding energies. Additionally, the authors should show the Cr 2p spectra to show if the Cr content is truly in form of CrN and not partially metallic Cr (as the EMPA results suggest)
10) SEM results. In Figure 5 what is the substrate? Fig.5e is annoyingly out of focus, and this is unfortunate as much of the conclusions are based on the changes on properties exactly with this coating deposited with -100V bias. Could the authors provide a better quality cross-section image of this sample?
11) 3.2 Mechanical properties. Could the authors discuss the hardness results a bit more and compare their results with the existing literature? How does this compare to other CrN-based coatings?
12) The dependency of the friction coefficient of the coating on the Si content was explained by the formation of the silicon hydroxide species on the surface. This should be visible in the XPS spectra too as the technique is really sensitive to exactly these kind of surface species. the Fig. 3 and the discussion, however, do not mention anything about the surface oxide (-or hydroxide) formation and it's difference in samples with 0V or higher bias voltage. So does any data support the speculation of the hydroxide formation or not?
Reviewer 2 Report
The manuscript “ Effects of bias voltages on the structural, mechanical and oxidation resistance properties of Cr-Si-N nanocomposite coatings” deals with the deposition and characterization of Cr-Si_N coatings by multi arc ion plating, using the bias voltage as main control parameter.
The manuscript is generally poorly written, with many English errors and expressions that are difficult to follow. Moreover, some of the explanations are quite superficial and do not take into account the full complexity of the phenomena being analyzed. The experimental results seem to be reasonable however, but they lack proper explanations and interpretations. The manuscript cannot be accepte for publishing as it is, a major revision is necessary.
Besides the general comments, here are few hints on some points to improve. Please note that English writing must be significantly improved, in the following you will find only some of the most striking examples:
Abstract:
Line 9 please change “ bias voltage on component” with “ bias voltage on composition”
Line 10 please change “were detailed investigated” with “were investigated in detail”
Line 10 to 12, please rephrase
Line 14 please change changed from loose to fine, compact to coarsening’, it is no clear what it is meant
Line 15 it is said“ micro-hardness of the coatings increased first and then decreased”. Please rephrase avoiding the use of “first”. It induces a perception about temporal evolution and it is not the case.
Line 19-20 please rephrase “had a significant effects”
Introduction:
Line 28, rephrase “coating protective material”
Line 32-33 the formulation “ According to the idea of nanocomposite coating proposed by Veprek” is to general, please specify the content of the “idea”
Line 36 “ and other comprehensive properties”, use of comprehensive is improper
Line 39-40, please avoid multiple use of “process”
Line 40. Please rephrase “process of coating”, incorrect English topic
Line 40-42 please explain what “a large number of plasma” means. It is very unusual to formulate like this. Moreover the enumeration of plasma constituents is incomplete
Line 44, please rephrase “the application of bias voltage makes particles bombard the coating”
In the last paragraph of introduction there are information about the existing studies that are misleading. The use of proper bias and effect of bias voltage is investigated in many studies. Please support your affirmations with references. There are studies, maybe not in the exact system or for the exact material, but all this has to be supported by references.
Experimental details:
Line 65 please correct “were firstly ultrasonic cleaned in” and “then followed by ultrasonic cleaned in”
Line 78 please rephrase “the coating time was controlled at 50 min”
Please provide more details about the experimental setup, geometry, type of vacuum vessel, powers supplies, electrical characteristics etc
Characterisation:
Different size and font can be found, please correct
Line 86 the current is expressed in W. please correct.
Please specify what 0 bias voltage means. Was the substrate floating or grounded?
In the explanation of Si content there are a few other phenomena that are not taken into account: the different ionization degrees of the elements involved, the changes in plasma due to the presence of bias voltage (when changing from 0 to -50 V).
Lines 120-122, the explanation given related to the change of growth direction is only an assumption. It is not clear how it is related with results from the cited reference. Was the stress evaluated in any way?
Please give more detiles about the peak fitting in XPS. The peak at 397.9 does not seem very visible
Line 149, it is written “under different bias voltages of -100 V.”
Line 160-163 the number and size of droplets is discussed in relation to bias voltage. Are the observations statistically relevant or just a choice of image? provide average values of roughness to confirm that the trends are consistent
Line 168-172. There are some explanations regarding the droplets distribution, saying that they are “bombarded off”. It is not very probable that ions can remove large dropplets from surface just by knocking them out. More probably it is related to other phenomena such as larger stress of the coatings that support the droplets, higher energy of dropplets and smaller sticking coeffect, resputtering, heating or surface charge accumulation etc. please add more plausible explanations
Please differentiate between particles, as in droplets, and particles, as in atoms, ions. It is not very clear in the discussions
Line 178, please rephrase “the grain size is refined well”
Please do not use past tense when talking about content of figures.
Line 190 to 193, please reconsider this explanation. The higher energy of ions does not increase the number of ions. The increased flux would do so.
Please give details about how the author measured the hardness and the thicknes. The relation between density and thickness is not clear, since higher density would mean lower deposition rate, at equivalent material content.
Line 211, pleae rephrase, avoid using “obvious”
The explanation on friction coefficient variation is centered on Si content only. Please consider other important factors that can play an importn role: hardness of the coating, surface roughness, number of dropplets on the surface and possible formation of a tribolayer from their residues.
Line 241, please rephrase “were completely disappeared”
Fig 10, please add information on the content in the figure caption. Assign A, b, c etc to different conditions
Conclusions
Line 269, avoid using “refined well” and “first increased” line 272
Please add to the conclusions anu relevant explanation resulting from changing the text as suggested.
Reviewer 3 Report
Dear Authors,
After analyzing the text of your submissions, I find that you are taking on an important and recent topic that is worth publishing in the Coatings journal. I believe you should make some changes to the manuscript. I list them in the order of their appearance in the text:
Abstract: Please add quantitative results.
The introduction is quite short, I suggest you expand on it using articles published by MDPI publishing house. It will also allow you to increase the number of references, the number of which for an article of this type should be around 25-30.
Please cite the references according to the author guidelines (not superscript).
Line 55: cite the sources mentioned at this point.
Line 63: Please describe the materials in more detail.
Lines 68 and further: add a space before degrees C.
Please pay close attention to the formatting of the text. The font size is not right in many places: 81, 118, 122-13 ...
Line 86: change "ev" to "eV". Check the notation of the parentheses.
Lines 90, 91: No need to underline the text.
Figures should be signed as: "Figure" and not as abbreviation: "Fig.".
Figure 1: in my opinion the axis signatures should be: Elements ’content and Bias voltage.
Figure 4 should be after line 155. Figure 5 should be on line 186.
Figure 5: photos overlap caption.
Figure 7: change "Hv" to "HV".
Figure 9 should be on line 249.
Lines 250-261: change font size.
References must be formatted according to the guidelines and supplemented with current (last 2 years) scientific articles.
Round 2
Reviewer 1 Report
The authors have done a sufficient job in addressing the reviewers' comments and improving the revised manuscript. Therefore, I can recommend the Editor to accept this manuscript to be published in Coatings.
Reviewer 2 Report
The authors of the manuscript “ Effects of bias voltages on the structural, mechanical and oxidation resistance properties of Cr-Si-N nanocomposite coatings” responded to most of the issues raised during the review process. As for the English language, I would suggest them to get some help from native speaker or to use a professional proofreading service to improve the quality of the manuscript.
Moreover, in the text referring t the roughness please add the unit (line 172)
